# A Prospective Comparative Analysis Regarding the Assessment of Oral Mucosal Disease Using a Validated Questionnaire to Improve the Teaching of Dental Students

**DOI:** 10.3390/ijerph19159663

**Published:** 2022-08-05

**Authors:** Nico Roeschmann, Andrea Rau, Marco R. Kesting, Eva Maier, Mayte Buchbender

**Affiliations:** 1Department of Oral and Maxillofacial Surgery, Friedrich-Alexander-Universität Erlangen-Nürnberg, Glückstraße 11, 91054 Erlangen, Germany; 2Department of Oral and Maxillofacial Surgery, University of Greifswald, 17475 Greifswald, Germany; 3Dental Clinic 1—Operative Dentistry and Periodontology, Friedrich-Alexander Universität Erlangen-Nürnberg, Glückstrasse 11, 91054 Erlangen, Germany

**Keywords:** dental students, oral health, oral mucosal lesions, teaching strategies

## Abstract

Structured examination and treatment are essential in medicine. For dental students, a structured approach to the assessment of oral mucosal lesions is missing thus far. To validate an approach, a structured questionnaire was compared with the habitually used free description of oral lesions (white lesions, ulcers, hyperplasia). Thirty-three dental students were divided into two groups (Group 1 (*n* = 17) used the free description; Group 2 (*n* = 16) used a guided questionnaire) to characterize mucosal lesions in patients and make a tentative diagnosis. Although no difference was found between the groups regarding the suspected diagnosis or the histopathological findings, there was a significant advantage of the structured questionnaire in all aspects of the description compared to the free description (*p* = 0.000018). Thus, a structured description is an important aspect in the evaluation of oral mucosal changes, and a guided questionnaire should be implemented in the study of dentistry.

## 1. Introduction

The dentist’s field of activity comprises more than the treatment of the tooth and its disease; it includes the whole oral cavity, specifically the oral mucosa. This is clarified by using the term oral medicine, which is associated with dental medicine. Changes in the oral mucosa occur in more than 1 out of 4 in the population aged 17 and older, and in 1 out of 10 children between 2 and 17 years of age [1,2]. In Germany, 7493 patients developed cancer of the oral cavity in 2017; when the pharynx is included, the total number is 14,150 [3]. The vast majority of diagnoses are of squamous cell carcinoma [4,5,6]. Even early lesions of squamous cell carcinoma are usually assessable by visual examination and without the need for other diagnostic tools [7]. However, histological confirmation of suspicious findings is always necessary [8,9]. Nevertheless, the incidence of oral squamous cell carcinoma continues to rise worldwide, with an estimated incidence of 377,713 new cases/year in 2020 [10]. A clinically tumor-suspicious lesion should be biopsied within a 14-day period from the time of detection as, according to the guidelines, a worsening of the prognosis due to any delay has been reported [11,12]. An early and correct diagnosis is crucial for the prognosis as well as the patient’s subsequent quality of life [11,12,13,14,15]. However, since there are a large number of possible changes on the oral mucosa, which cannot always be classified as benign or malignant purely by clinical visual diagnosis, a uniform designation must serve. In addition, there are changes that cannot be clearly delineated as benign or malignant but can also be based on other etiologies such as infectious, reactive or immune-mediated, for example. Furthermore, there are also changes that are considered potentially malignant but do not yet meet the clinical or pathological criteria of an invasive malignant tumor but, strictly speaking, are not benign. These are, for example, changes such as leukoplakia or erythroplakia. Therefore, the definition of Oral Potentially Malignant Disorders (OPMDs) has been established by the WHO and includes a wide spectrum of changes, namely and currently updated as follows: leukoplakia, erythroplakia, proliferative verrucous leukoplakia, oral lichen planus, oral submucous fibrosis, palatal lesions in reverse smokers, lupus erythematosus, dyskeratosis congenita, oral lichenoid lesions, and chronic graft-versus-host-disease [16].

Since the dentist is usually the only one who regularly intensively examines the oral cavity, including the mucosa, the early detection of changes is one of the most important tasks and places the dentist in a central position in the prophylaxis of oral mucosal diseases [17].

An appropriate and standardized concept regarding the assessment of the oral mucosa is missing in undergraduate dental curricula. This leads to uncertainty among students [18,19,20]. In particular, potentially malignant disorders are often misjudged [18,21]. A Brazilian study compared three questionnaires to describe photographs of oral mucosal lesions and showed that questionnaires with schematic illustrations could increase the quality of the information transmitted [22]. Moreover, this increase supports the progression of interdisciplinary digital exchange and can avoid possible errors that can lead to a delay in treatment, such as misinterpretation of the classification into benign or malignant findings [22]. Therefore, not only is a structured procedure in terms of tooth treatment useful, but it should also be essential for examining the oral mucosa and should be an integral part of the preventive dental examination.

To the best of our knowledge, there is no established procedure for the assessment of various changes in the oral mucosa in the current literature in the field of dentistry. It can be assumed that the assessment of an oral mucosal lesion is almost exclusively supported by empirical values. Therefore, this study aimed to compare a validated questionnaire and its practical application to patients, in comparison to conventional free text documentation in dentistry studies, to establish a standardized assessment of oral lesions. We hypothesized that defining a diagnosis would be more predictable using the validated questionnaire.

## 2. Experimental Section

### 2.1. Study Design and Setup

This prospective comparative study was conducted at the Friedrich-Alexander-Universität Erlangen-Nürnberg, in the Department of Oral and Maxillofacial Surgery, from October 2018 to February 2020. It was approved by the ethics committee of the Friedrich-Alexander-Universität Erlangen-Nürnberg (FAU) on 17 July 2018 (149_18B).

All dentistry students who had completed their fourth year of study were asked to participate in the study as examiners. After signing a written informed consent form, those willing to participate were randomly divided into two groups (Group 1 = standard group and Group 2 = experimental group). All students in Groups 1 and 2 were assigned a patient with an oral mucosal lesion (white lesion, ulcer or hyperplasia). Patients of all sexes and ages (except minors and pregnant patients) were included. Further exclusion criteria for patients were previous biopsies in the same region and confirmed oral squamous cell carcinoma in the same or different location in the mouth, pharynx or somewhere else. All included patients also signed written informed consent forms.

The task for all students in both groups comprised the following four steps:

To collect the medical history of the patient, to perform a standardized intraoral investigation, to give a detailed description of a mucosal lesion and, finally, to come to a conclusion with a suspected diagnosis. For the description of the mucosal lesion, students either followed the standard protocol (Group 1), which included a template for a structured medical history and only a blank field in the description section, or it used a detailed questionnaire (Group 2), which was inspired by the questionnaire of Zimmermann et al. [22] and contained additional graphics and further tools, as seen in the Supplementary Materials. Each student was given 10 min of processing time.

Afterward, the lesion was photographed and a biopsy was taken by the same oral surgeon. The lesion selected for biopsy was also the one detected or assessed by the student. Therefore, only one lesion was examined histologically. The surgeon did not comment on the lesion in the presence of the students or patients and was not aware of the students’ diagnosis. The photographs were reviewed by a panel of experts, and a pattern description and suspected diagnosis for each lesion was determined as a master definition.

### 2.2. Design of the Questionnaire for Group 2

The questionnaire based on Zimmermann et al. [22] shows illustrations to define the lesion and includes questions about the patient’s medical history or tobacco use, HIV status, diabetes, and skin diseases (Figure 1 and Figure 2). Moreover, it includes listed characteristics such as localization, size, type of lesion, surface, and margin conditions.

### 2.3. Data Extraction and Examination

A scoring system was applied for the categories of diagnosis and description.

For the category of diagnosis, a maximum score of 2 points could be achieved—one point for making a suspected diagnosis, and an additional point if the suspected diagnosis matched the histopathological findings of the biopsy.

For the category of description, the data collected from the questionnaires were assessed in the following categories, which are considered characteristic and important diagnostic criteria, according to the panel of experts:Localization.Size.Type of lesion (according to the inclusion criteria).Surface and margin condition.

The evaluation of the criteria marked in the questionnaires was based on a point system, the details of which are displayed in Table 1. In total, a score of 9 points could be achieved. One point could be scored for each description in the categories. All descriptions were compared with those of the experts and, in contrast to Zimmermann et al., a category was scored if it was just described, even if it was different from the experts’ description. The categories “Localization”, “Size”, and “Type of lesion” each contained one description. The category “Surface and margin condition” contained four descriptions, where one point could be scored for each description. An additional point could be scored in the categories “Size” and “Type of lesion” if the description was similar to the experts’ or when the necessary information was provided.

### 2.4. Outcomes

The primary outcome was defined as the agreement of the stated suspected diagnosis with the actual pathological diagnosis. The secondary outcome was defined as the difference between the groups in terms of the students’ descriptions.

### 2.5. Data and Statistical Analysis

SPSS software, version 24 (IBM, Armonk, New York, NY, USA), was used for statistical analysis. For testing for a normal distribution, the Shapiro–Wilk test was used, and the significance level was set at *p* < 0.05. For the elaboration of differences, Fisher’s exact test was used for the secondary outcome category “Localization”, and the Mann–Whitney U test was used in the primary outcome for diagnosis and in the secondary outcome categories “Size”, “Type of lesion” and “Surface and margin condition”, with a significance level of *p* < 0.05. The results are also presented in mean ranks. Higher ranks mean more points were achieved.

## 3. Results

### 3.1. Participant Cohorts

A total of *n* = 33 patients and *n* = 33 students participated. In total, *n* = 17 male and *n* = 16 female patients participated with an age range from 20 to 84 years. Regarding the students, Group 1 (*n* = 17) included *n* = 6 male and *n* = 11 female students with a mean age of 24.29 (±1.49). Group 2 (*n* = 16) included *n* = 4 male and *n* = 12 female students with a mean age of 24.31 (±2.11). The distribution of students and the definitive pathological diagnosis of patients are shown in Table 2.

### 3.2. Primary Outcome—Diagnosis

Group 1 achieved a mean rank of 15.47 (Figure 3). Three students scored 0 points because no suspected diagnosis was given; *n* = 9 students scored 1 point; and *n* = 5 students correctly matched their suspected diagnosis with the pathological diagnosis. The suspected diagnoses were squamous cell carcinoma (*n* = 6), and oral lichen planus (*n* = 4).

Group 2 achieved a mean rank of 18.63 (Figure 4). All students in this group made a suspected diagnosis, *n* = 10 students scored 1 point, and *n* = 6 students correctly matched their suspected diagnosis with the pathological diagnosis. The 3 most common suspected diagnoses were oral leukoplakia (*n* = 4), oral lichen planus (*n* = 4) and oral lichen erosivus (*n* = 2). Group 2 achieved slightly higher, nonsignificant scores (*p* = 0.367; *p* > 0.05).

### 3.3. Secondary Outcome—Description

For the descriptions in the 4 categories, a maximum score of 9 points could be achieved. In each group, one student achieved the maximum score. Three students in Group 1 scored 0 points. In total, Group 1 achieved a mean rank of 10.56, and Group 2 achieved a mean rank of 23.84. The result was statistically significant, with *p* = 0.000018 (Figure 5).

Regarding each category individually, more significant differences could be found. The description of “Localization” was made in 70.6% of cases in Group 1 and in 100% of cases in Group 2; the difference was significant, with *p* = 0.044. In the second category, “Size”, 64.7% of Group 1 did not describe the size. In Group 2, the size was not described by 12.5% of the students. Therefore, Group 1 reached a mean rank of 12.47, and Group 2 reached a mean rank of 21.81, which was statistically significant (*p* = 0.004). The “Type of lesion” was described correctly by 23.5% of Group 1 and by 50% of Group 2. The mean ranks of Group 1 (12.82) and Group 2 (21.44) showed a statistical significance (*p* = 0.005). The maximum score in the category “Surface and margin conditions” was achieved by three students, one in Group 1 and two in Group 2. In total, the mean ranks in the last category were 13.29 for Group 1 and 20.94 for Group 2, with a statistical significance (*p* = 0.019).

Examining how often students failed to provide information in one of the four categories, Group 1 was missing information in 44.1% of cases; in Group 2, information was missed in 6.3% of cases.

## 4. Discussion

A visit to the dentist should include not only an examination of the teeth, but also a standardized assessment of the entire oral cavity and the oral mucosa. To live up to the responsibility of being the specialist who regularly examines patients’ oral mucosa, expertise in this diagnostic field is essential for dentists. Due to the different anatomy of the various tissues, (i.e., keratinized attached gingival tissue, mucosal nonattached tissue, the tongue, and pharynx), there are a variety of mucosal lesions, and differentiating between benign and malignant lesions can be challenging. In general, benign/malignant changes of the oral mucosa are correctly diagnosed in 80.7% of the cases [9]. In the study of Patel et al., it was only 63% of the cases. Thus, benign changes are clinically classified as such in 97% of the cases, and malignant changes are correctly classified in 57% [14].

Considering our results, the correct (benign or malignant) diagnosis was made in 33% of the cases. Regarding the study of Cerero et al., 692 students from study years 3, 4 and 5 had to classify 40 photographs of oral mucosal changes into one of the 3 categories, benign, malignant or OPMDs. On average, 42.8% of the changes were correctly classified [18]. Similarly, in the study by Gaballah et al., 80 5th-year students assessed 32 photographs of oral mucosal changes and had to classify them into 4 categories (normal variation, benign, OPMD, and malignant). The students chose the correct answers in 44.1% of the cases [21]. Since, in our study, the diagnoses had to be formulated by the students themselves and thus a higher degree of difficulty has to be assumed, this could explain the difference in the success rate.

However, no significant difference was found between the groups in determining the diagnosis according to definitive pathology. Information about a suspected diagnosis was missing in 44.1% of the cases. From this, it is clear that students have difficulties in the assessment and classification of oral lesions. This finding is in accordance with the study of Hassona et al., in which 456 students were interviewed about their satisfaction with their current teaching. In this study, 88.7% of the students claimed that the training to diagnose oral lesions was insufficient [24]. Similarly, in the study by Keser et al., 80.4% of students felt they had not received sufficient training to perform an oral cancer examination [19].

Regarding the description of the lesion, Group 2 provided significantly more information about the lesion, even when the diagnosis was not in accordance with the pathology. A reason for that might be that the students were not theoretically aware of possible changes and its characteristics, benign or malignant, at the time of the survey and therefore no correct diagnosis could be derived. Nevertheless, theoretical knowledge is the foundation of correct assessment [24]. Although students report uncertainty, studies clearly show that they have the necessary knowledge. In Keser et al., 3rd-year students knew 83.3% of the important risk factors, compared to 91.4% in the 5th year [19]. In the study of Srivastava et al., 100% of students reported feeling well informed about the clinical appearance of oral cancer. However, 69.7% of the same students reported insufficient knowledge about the prevention and detection of oral cancer [20]. This discrepancy between existing theoretical knowledge and practical applications might be reduced by using a structured questionnaire. There is currently no established concept in teaching that enables dental students to be trained at an early stage. Zimmermann et al. showed that a structured approach is advantageous in the description of images of oral mucosa changes [22]. In the current literature, no other studies have evaluated training approaches for dental students in terms of recognizing oral lesions.

Thus, the integration of a standardized questionnaire seems to be a useful approach to improve teaching. This is shown in the category of the description, which is essential to define a suspected diagnosis, where significantly higher scores were achieved compared to the group with the standardized questionnaire.

The detailed assessment of the characteristics rated as important for mucosal changes in this study, in Group 1 compared to Group 2, also shows that students in Group 2 were able to achieve higher scores. Thus, the exact description of the size, localization, type of lesion and surface and margin condition is essential for the diagnosis and classification of whether a lesion is benign or malignant. While the aspects of size and location were forgotten in Group 1 due to a lack of a structured approach, the images obviously facilitated the classification of the type of lesion and the surface and margin condition in Group 2 considerably. This is consistent with the findings of Frola and Barrios. In their study, 121 dental students were asked to describe oral changes associated with oral cancer. While most were aware of induration and a prolonged persistence of the lesion for more than 15 days, very few could name abnormalities such as an irregular edge (10%) or changes in oral texture (>15%) [25]. These results showed that many students are not aware of how to derive an appropriate diagnosis based on their clinical findings. Despite having theoretical knowledge, translation of this knowledge into practical applications seems to be the problem. By using a structured questionnaire, remembering to document important aspects becomes more likely. Furthermore, the use of a questionnaire may also ease interprofessional communication in online formats, for example, as in COVID-19 pandemic times. Furthermore, telemedicine applications could also benefit from such an approach [26,27].

Pei & Wu show that a change to online teaching does not have to have a negative impact on learning success. In their meta-analysis, none of the studies showed a disadvantage compared to the offline method. Among other things, it is important to adapt to students’ preferences and characteristics [28]. Different learning concepts show significant advantages in learning performance [19,29,30,31]. These new learning methods and concepts are necessary to meet the new requirements of the license to practice medicine. An increased link to oral medicine and systemic aspects is explicitly mentioned for dental students [32]. Moreover, a standard procedure and description of a lesion leads to a uniform interdisciplinary approach. This means that everyone knows exactly what is being discussed when a decision has to be made. In this way, misunderstandings and, in the worst case, wrong decisions can be avoided, and the treatment of possible malignant diseases can begin more quickly. Therefore, practitioners would also benefit from a standardized procedure.

There are a few shortcomings in this study that should be mentioned and discussed. Only a limited spectrum of oral lesions was assessed by the students. Some might have been easier to diagnose than others. Only one lesion was described by each student whereas not all OPMDs are focal or isolated. Furthermore, the derived evidence from this study must be considered weak due to the small number of students. Additional studies with this questionnaire should compare clinicians with more or less experience and include more students. Additionally, a comparison of students from different study years with prior structured theoretical training might be interesting.

## 5. Conclusions

Although no difference was found between the groups regarding the suspected diagnosis or the histopathological findings, there was a significant advantage of the structured questionnaire in all aspects of the description compared to the free description. By using standardized questionnaires, students are able to provide better descriptions and therefore pay more attention to details of oral mucosal lesions—a prerequisite for the early diagnosis and treatment of potential malignant disorders.

## Figures and Tables

**Figure 1 ijerph-19-09663-f001:**
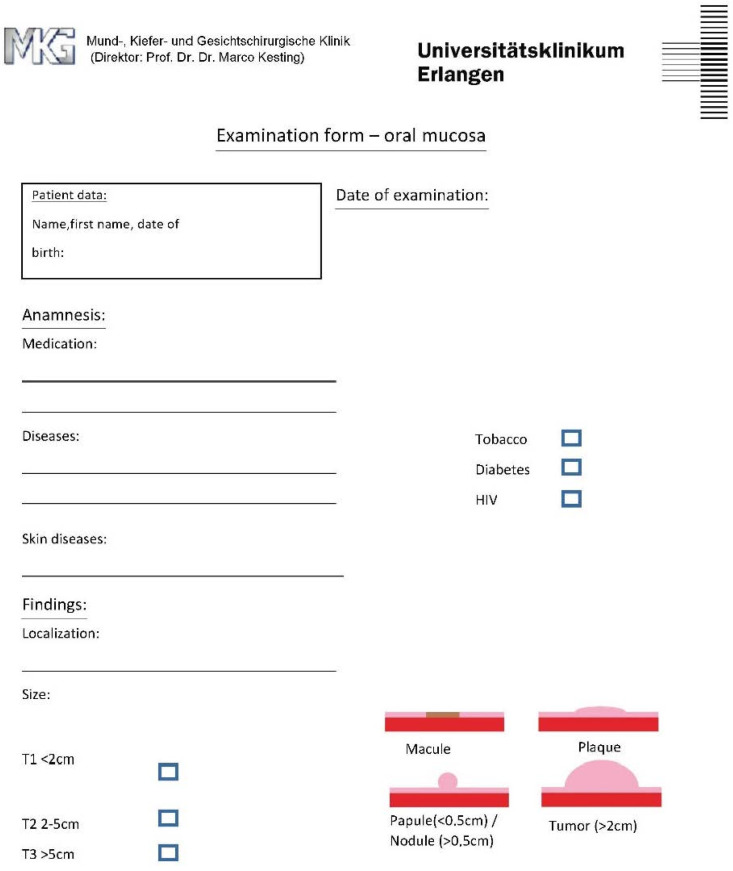
First page questionnaire for Group 2 inspired by [22,23].

**Figure 2 ijerph-19-09663-f002:**
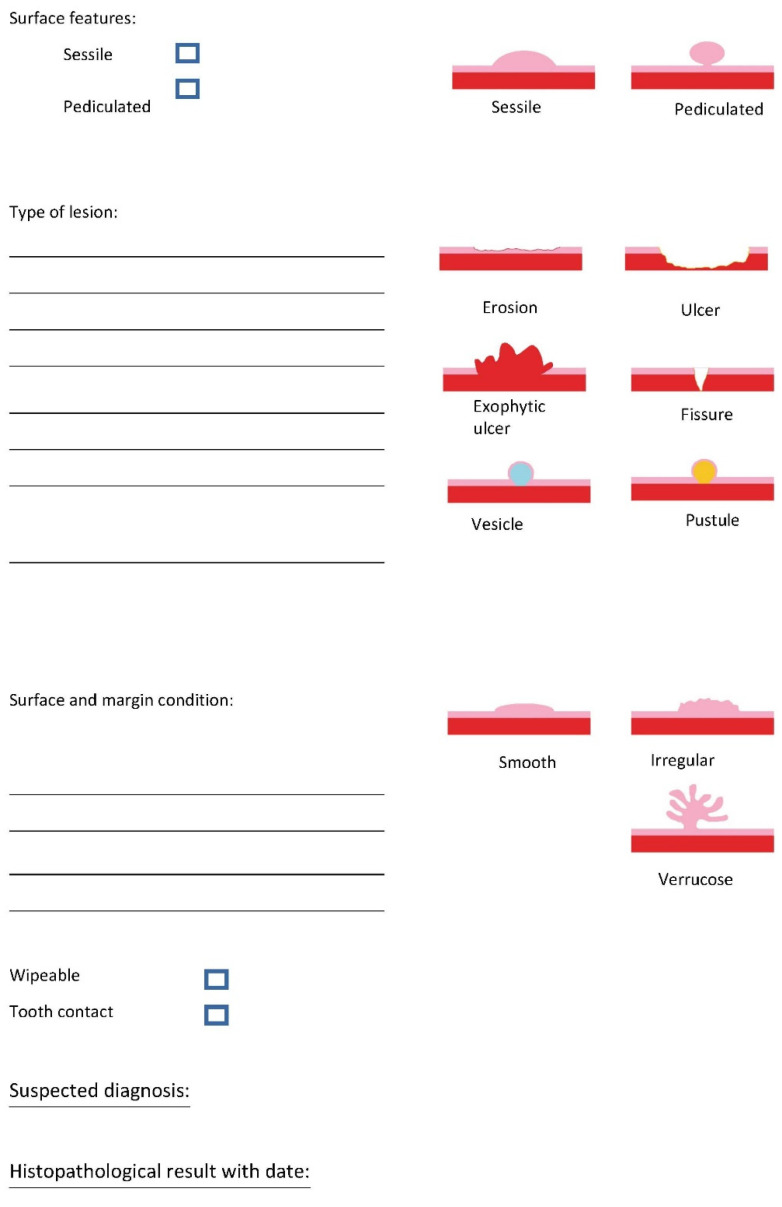
Second page questionnaire for Group 2 inspired by [22,23].

**Figure 3 ijerph-19-09663-f003:**
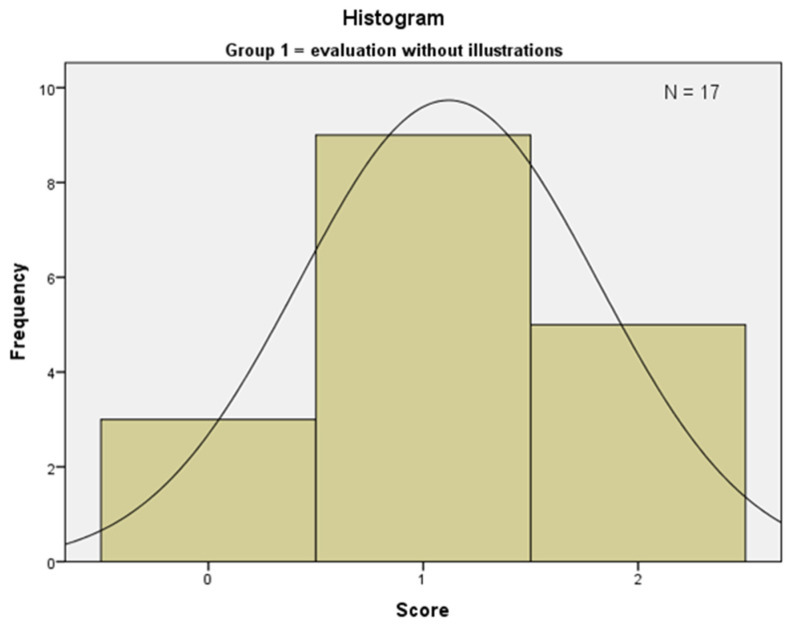
Showing the distribution of the achieved scores of Group 1.

**Figure 4 ijerph-19-09663-f004:**
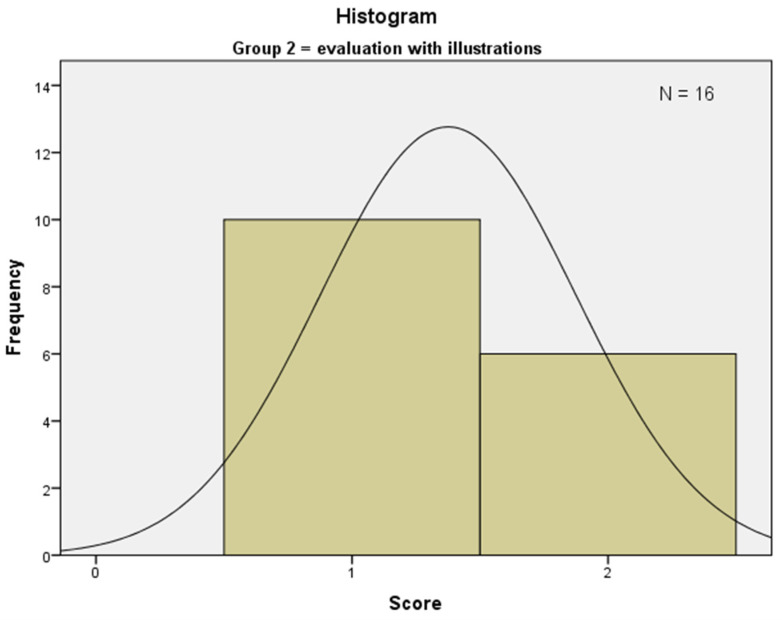
Showing the distribution of the achieved scores of Group 2.

**Figure 5 ijerph-19-09663-f005:**
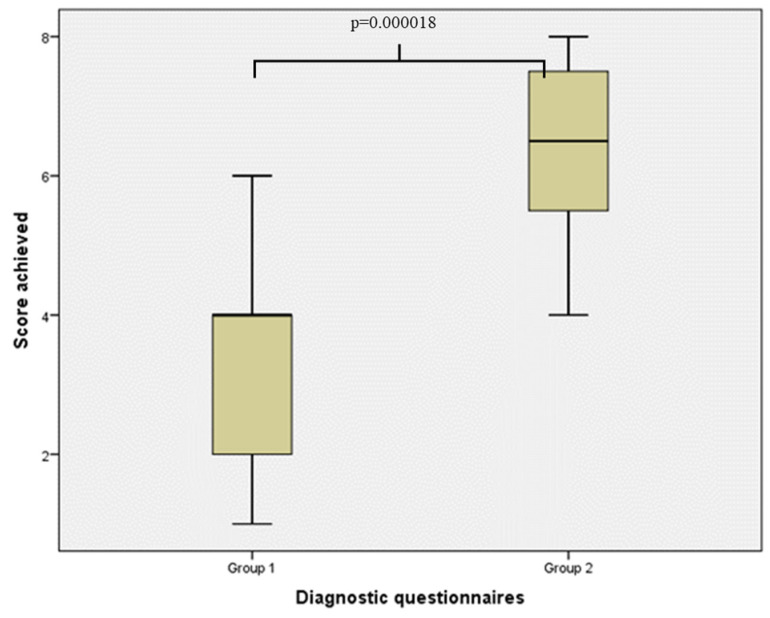
Comparison between Group 1 and Group 2 in terms of findings in all four categories: Localization, Size, Type of lesion, Surface and margin conditions. The use of the Mann-Whitney U-test shows significance *p* = 0.000018.

**Table 1 ijerph-19-09663-t001:** Category and related maximum scores.

Category	Scoring System	Maximum Score
Localization	1 point for description	1
Size	1 point for description	2
	1 point for classification	
Type of lesion	1 point for description	2
	1 point for classification	
Surface and margin condition	1 point for color description	4
	1 point for margin description	
	1 point for surface description	
	1 point for shape description	
Total		9

**Table 2 ijerph-19-09663-t002:** Distribution of students, students’ diagnosis and patients’ correlated diagnosis.

Patient Number	Group	Students Diagnosis	Pathological Diagnosis
1	1	None	Aphtous stomatitis
2	2	Aphtous stomatitis	Oral lichen planus
3	2	Oral lichen planus	Oral leukoplakia
4	2	Oral lichen planus	Oral herpes simplex
5	1	Oral lichen planus	Oral herpes simplex
6	1	Oral lichen planus	Oral lichen planus
7	1	Oral squamous cell carcinoma	Oral leukoplakia
8	2	Oral leukoplakia	Oral leukoplakia
9	1	Fibrous tumor	Irritation fibroma
10	1	Oral leukoplakia	Oral herpes simplex
11	2	Oral lichen planus	Oral lichen planus
12	1	Oral squamous cell carcinoma	Oral leukoplakia
13	2	Oral lichen erosivus	Oral squamous cell carcinoma
14	1	Oral candidiasis	Oral leukoplakia
15	1	Oral leukoplakia	Oral leukoplakia
16	1	Oral lichen planus	Oral lichen planus
17	2	Oral leukoplakia	Oral lichen planus
18	1	Oral lichen planus	Oral graft-versus-host disease
19	2	Oral lichen erosivus	Oral lichen erosivus
20	1	Oral squamous cell carcinoma	Oral squamous cell carcinoma
21	2	Oral leukoplakia	Oral leukoplakia
22	1	None	Oral leukoplakia
23	2	Acute necrotizing ulcerative periodontitis	Oral lichen planus
24	1	Oral squamous cell carcinoma	Trauma of oral mucosa
25	1	Oral leukoplakia	Oral lichen planus
26	2	Oral squamous cell carcinoma	Oral squamous cell carcinoma
27	1	Oral squamous cell carcinoma	Oral squamous cell carcinoma
28	2	Oral leukoplakia	Oral squamous cell carcinoma
29	1	None	Oral lichen planus
30	1	Oral squamous cell carcinoma	Oral leukoplakia
31	2	Oral lichen planus	Oral lichen planus
32	2	Lupus erythematosus	Oral lichen planus
33	1	Oral erythroplakia	Oral lichen planus

## Data Availability

The data presented in this study are available on request from the corresponding author.

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
