# Peer review of "A Prospective Comparative Analysis Regarding the Assessment of Oral Mucosal Disease Using a Validated Questionnaire to Improve the Teaching of Dental Students"

_ijerph, 2022, doi:10.3390/ijerph19159663_

Round 1

Reviewer 1 Report

This manuscript is a well-written and structured article, addressing the importance of oral disease diagnosis through a structured questionnaire, focusing on the medicine teaching framework. My questions and recommendations are the following:

Regarding the sample size that determined the student/patient relationship, indicate the rationale for its determination, because the fact that two students, one from each group, did not see each patient, but only a student from one of the groups; that may have been a limitation as well, in the sense of having possibly reduced some power from the applied statistical comparative tests

Is it possible to report in the manuscript an idea of the average time it took to complete the forms in each group and/or if the students had a specific time limit to perform it?

Regarding the patients, indicate how they were distributed for each group of students and characterize, if possible, what oral pathologies they had effectively.

Author Response

Dear Reviewer,

On behalf of my fellow co-authors, I would like to thank you for your esteemed effort and time for reviewing our manuscript. We have edited and improved the first draft and thus hope to have improved the manuscript. We thank the reviewer for accepting to re-read the revised manuscript.

This manuscript is a well-written and structured article, addressing the importance of oral disease diagnosis through a structured questionnaire, focusing on the medicine teaching framework. My questions and recommendations are the following:

Point 1: Regarding the sample size that determined the student/patient relationship, indicate the rationale for its determination, because the fact that two students, one from each group, did not see each patient, but only a student from one of the groups; that may have been a limitation as well, in the sense of having possibly reduced some power from the applied statistical comparative tests

Response 1: This is a correct and important point, thank you for your remark. Based on our analysis before, two groups should be compared separately so that only the possible change in assessment by the questionnaire is examined first and not its’ impact on the assessment of different changes. Of course, for all possible changes (e.g. also pigmented lesions etc.) it would be useful in the future that the questionnaires are also examined for this variety of changes in their significance. This will be planned for follow-up studies.

Point 2: Is it possible to report in the manuscript an idea of the average time it took to complete the forms in each group and/or if the students had a specific time limit to perform it?

Response 2: Yes, of course. We added this information in the methods section (Line 99). Unfortunately, we did not measure actual utilization in terms of time.

Point 3: Regarding the patients, indicate how they were distributed for each group of students and characterize, if possible, what oral pathologies they had effectively.

Response 3: Thanks for the remark. The patients were randomly distributed to the students’ groups. We added a Table 2 in the results section (Line 152)were the definitive pathologies are marked. We think that this substantially improved this manuscript.

Reviewer 2 Report

Comments to the author for manuscript entitled: "A prospective comparative analysis regarding the assessment of oral mucosal disease using a validated questionnaire to improve the teaching of dental students".

The manuscript is interesting and relevant. It is of great importance to teach dental students to pay attention to oral mucosal lesions and how to describe and document lesions correctly in order to establish accurate differential diagnoses. This type of documentation is important both for the purpose of monitoring the patient (identifying changes over time), for better communication between different health-care providers and also for research purposes.

The text is clear and easy to read.

As the authors point out the importance of establishing a standardized assessment of oral lesions is especially important in pre-malignant lesions and disorders. Thus, the introduction should further refer to the term Oral Potentially Malignant Disorders according to the classification and nomenclature suggested by the WHO Collaborating Centre for Oral Cancer (Warnakulasuriya et al. Oral Dis. 2021 Nov;27(8):1862-1880. doi: 10.1111/odi.13704.).

 Larger group of patients with wider range of types of lesions could be more contributary for assessing the usefulness of the structured questionnaire. 

In cases of extensive or multifocal lesions it is not clear by what criteria the site was selected for performing the diagnostic biopsy. In extensive or multifocal OPMDs diverse microscopic findings may be present at different locations and that could disrupt the correlation between clinical and microscopic diagnosis. Although the description in MM (lines 76-77) indicates that the selected lesions were isolated (focal), the results (lines 144-151) include several cases of Oral Lichen Planus, which by definition are not focal or isolated lesions.

Line 140: the number of participants should be written in the same way as in the other sentences.

Several comments regarding the questionnaire:

·       It should also address alcohol consumption habits.

·       The questionnaire should address whether the suspected lesion is focal or multifocal. In some OPMDs, but also in other pathological conditions (i.e., pigmented lesions) this parameter has a great diagnostic significance. For example, it is essential for the distinction between Leukoplakia and Proliferative Verrucous Leukoplakia.

·       The questionnaire should include description of the lesion on palpation (soft, indurated, fluctuant etc.). The decision about the potential nature of a lesion (for example, whether there is a risk of dysplastic changes or even malignancy in a white lesion) is based on parameters achieved by palpation and not only by appearance.

Author Response

Dear Reviewer,

On behalf of my fellow co-authors, I would like to thank you for your esteemed effort and time for reviewing our manuscript. We have edited and improved the first draft and thus hope to have improved the manuscript. We thank the reviewer for accepting to re-read the revised manuscript.

The manuscript is interesting and relevant. It is of great importance to teach dental students to pay attention to oral mucosal lesions and how to describe and document lesions correctly in order to establish accurate differential diagnoses. This type of documentation is important both for the purpose of monitoring the patient (identifying changes over time), for better communication between different health-care providers and also for research purposes.

The text is clear and easy to read.

Point 1: As the authors point out the importance of establishing a standardized assessment of oral lesions is especially important in pre-malignant lesions and disorders. Thus, the introduction should further refer to the term Oral Potentially Malignant Disorders according to the classification and nomenclature suggested by the WHO Collaborating Centre for Oral Cancer (Warnakulasuriya et al. Oral Dis. 2021 Nov;27(8):1862-1880. doi: 10.1111/odi.13704.).

Response 1: Thanks for the important remark. We added this term in the introduction section (Line 42-44).

Point 2: Larger group of patients with wider range of types of lesions could be more contributary for assessing the usefulness of the structured questionnaire.

Response 2: Yes, that is quite a valid point. Especially since the effect of such a questionnaire on the broad spectrum of changes could then also be investigated. However, in this study we aimed to work out the effect of such a questionnaire in comparison to no questionnaire. Even if the significance of this is of course reduced due to the small number of patients, which we also discussed in the shortcomings section.

Point 3: In cases of extensive or multifocal lesions it is not clear by what criteria the site was selected for performing the diagnostic biopsy. In extensive or multifocal OPMDs diverse microscopic findings may be present at different locations and that could disrupt the correlation between clinical and microscopic diagnosis. Although the description in MM (lines 76-77) indicates that the selected lesions were isolated (focal), the results (lines 144-151) include several cases of Oral Lichen Planus, which by definition are not focal or isolated lesions.

Response 3: This is correct and we have modified it in the MM part (98-98). The lesion selected for biopsy was also the one detected or assessed by the student. Therefore, only one lesion was examined histologically in the OLP. To make this fact clear we mentioned in the shortcoming sections now.

Point 4: Line 140: the number of participants should be written in the same way as in the other sentences.

Response 4: That’s right, we changed this.

Point 5: Several comments regarding the questionnaire:

It should also address alcohol consumption habits.

The questionnaire should address whether the suspected lesion is focal or multifocal. In some OPMDs, but also in other pathological conditions (i.e., pigmented lesions) this parameter has a great diagnostic significance. For example, it is essential for the distinction between Leukoplakia and Proliferative Verrucous Leukoplakia.

The questionnaire should include description of the lesion on palpation (soft, indurated, fluctuant etc.). The decision about the potential nature of a lesion (for example, whether there is a risk of dysplastic changes or even malignancy in a white lesion) is based on parameters achieved by palpation and not only by appearance.

Response 5: Thank you for these comments, which we will definitely accept to revise the questionnaire for our follow-up studies. These are certainly very important aspects that play a role in improving the assessment and early detection. Thank you very much.

Round 2

Reviewer 2 Report

Although I have recommended to refer to the term Oral Potentially Malignant Disorders as defined by the WHO, the context in which it was incorporated into the text is not clear. The authors should consider to re-write this part of the introduction (lines 42-48) in order to emphasize the challenge of diagnosing the variety of lesions in the oral cavity – malignant, potentially malignant, benign, immune-mediated, infectious, reactive, etc.

In Table no.2 consider adding also the students' diagnosis.

Author Response

Dear Reviewer,

On behalf of my fellow co-authors, I would like to thank you for your esteemed effort and time for reviewing our manuscript. We have edited and improved the second draft and thus hope to have improved the manuscript. We thank the reviewer for accepting to re-read the revised manuscript.

Point 1: Although I have recommended to refer to the term Oral Potentially Malignant Disorders as defined by the WHO, the context in which it was incorporated into the text is not clear. The authors should consider to re-write this part of the introduction (lines 42-48) in order to emphasize the challenge of diagnosing the variety of lesions in the oral cavity – malignant, potentially malignant, benign, immune-mediated, infectious, reactive, etc.

Response 1: We are sorry that we have not made this aspect clear so far, we hope that now with the revision of lines 43-56 we have clearly dealt with this point.

Point 2: In Table no.2 consider adding also the students' diagnosis.

Response 2: Thanks for the remark, this point seems reasonable and we changed the Table according to this.